# Depth-consistent Motion Blur Augmentation

## Abstract

Motion blur is a ubiquitous phenomenon commonly encountered in lightweight, handheld cameras. Addressing this degradation is essential for preserving visual fidelity and ensuring the robustness of vision models for scene understanding tasks. In the literature, robustness to motion blur has been generally treated like other degradations; this despite the complex space-variant nature of motion blur due to scene dynamics and its inherent dependence on scene geometry and depth. While some recent works addressing this issue have introduced space-variant blur due to scene dynamics, they fall back on space-invariant blurring to model camera egomotion which is imperfect. This work proposes an efficient methodology to generate space-variant depth-consistent blur to model camera egomotion by leveraging depth foundation models. We refer to our approach as Depth-consistent Motion Blur Augmentation (DMBA). To demonstrate the effectiveness of DMBA in improving robustness to realistic motion blur, we provide experiments for the tasks of semantic segmentation and self-supervised monocular depth estimation. We include results for standard networks on the Cityscapes dataset for semantic segmentation and the KITTI dataset for monocular depth estimation. We also illustrate the improved generalizability of our method to complex real-world scenes by evaluating on commonly used datasets GoPro and REDS that contain real motion blur.

## 1 Introduction

Traditional cameras were heavy and sensitive and often warranted minimal motion during long exposure times to capture sharp images. Technology has made modern cameras significantly more compact with shorter exposure times. However, lightweight cameras suffer from poor camera stability, which often results in camera egomotion blur. Modelling this blur is challenging since the camera trajectory traversed can be due to hand shake or due to camera on a moving vehicle or a combination of both. Motion blur caused by any of these reasons can lead to performance drop, especially at close quarters where the blurring is more significant. This can have serious consequences in safety-critical scenarios such as autonomous driving. While recent works on foundation models for various scene understanding tasks do report better robustness and generalizability across datasets, they are still limited by their large sizes and high inference times, rendering them unsuitable for deployment. Additionally, finetuning these larger models is difficult when data is limited. Hence, there is an imminent need to make lightweight models robust to motion blur for scene understanding tasks.

Some recent works investigating the robustness of deep-learning models to commonly occurring degradations (Michaelis et al., 2019; Kamann & Rother, 2020a; Hendrycks & Dietterich, 2019) have benchmarked standard models for their robustness to multiple severity levels of sixteen commonly occurring degradations. Motion blur is one of the degradations considered and the experiments are conducted for multiple scene understanding tasks. While (Hendrycks et al., 2019) and (Lee et al., 2020) propose augmentation strategies to improve generic robustness of models across all the sixteen degradations, each of these works has oversimplified the modeling of motion blur by assuming it to be uniform and spatially invariant. A recently proposed work (Aakanksha & Rajagopalan, 2023) attempts to introduce space-variant motion-blur by leveraging exisiting semantic segmentation annotations to localize the blur in a class-centric manner to ensures better modelling of motion blur caused by scene dynamics. However, akin to other approaches, they simplify camera egomotion blur by reducing it to a space-invariant model by convolving the entire image with a non-linear $2D$ motion blur kernel. Motion blur is a complex phenomenon that depends on both object and camera

motion, and is inherently depth-dependent. A convolutional over-simplification of the blur model, in particular when modeling camera egomotion, is relatively easy to realise, but is valid only when the objects are far enough for the scene to be assumed as planar. To generate realistic space-variant camera egomotion blur for a static scene, one needs to consider pixel-wise scene depth. Since every pixel can have a different blur kernel depending upon its depth and the camera motion, this becomes prohibitively expensive to compute, especially when considering higher resolution images.

Building on the observation that foundation models (Kirillov et al., 2023; Li et al., 2025; Yang et al., 2024), while highly generalizable across diverse natural scenes, are often impractical for direct deployment due to their large parameter sizes and high computational requirements, we propose to employ them to enhance the performance of smaller, more efficient networks. We make use of a depth estimation foundation model (Yang et al., 2024) to produce dense depth maps which we then use to propose an efficient Depth-consistent Motion Blur Augmentation (DMBA) strategy grounded in $3D$. We also propose a scheme to generate $3D$ motion trajectories which are then used along with the estimated depth maps to synthesize realistic camera egomotion blur using depth-aware warping. We experiment with two representative scene understanding tasks, namely, supervised semantic segmentation and self-supervised monocular depth estimation, and establish the effectiveness of our approach in improving performance for these tasks on real blurry images without any additional costs in terms of parameters or inference time.

Our main contributions are the following :

1. We propose a depth-consistent motion blur augmentation strategy to generate spatially-varying realistic motion blurred images from sharp images. Our approach leverages foundation models to obtain dense depth maps, making our motion blur augmentation strategy generalizable across a wide range of datasets.

2. We develop a $3D$ trajectory generation pipeline to generate camera motion paths to simulate camera motion blur.

3. We establish the generalizability of our approach across multiple scene understanding tasks by showing that it improves robustness across motion-blurred images for both semantic segmentation and monocular depth estimation.

4. While our model is trained only on synthetically generated data, the realistic nature of our augmentation enables it to perform well on real blur datasets such as GoPro and REDS.

## 2  RELATED WORKS

Multiple recent works have attempted to evaluate and increase the robustness of CNNs in various naturally-occurring degradations. (Li et al., 2022) used defocus blur to reduce the impact of irrelevant background information on semantic segmentation. (Vasiljevic et al., 2016) examined the impact of blur on image classification and semantic segmentation using VGG-16 and found that using defocus blur augmentation leads to significant improvements in robustness for classification task but not for segmentation. Subsequently, (Hendrycks & Dietterich, 2019) introduced the ImageNet-C dataset where the authors corrupt the ImageNet dataset by common image corruptions and benchmark various pre-trained models for robustness. (Kamann & Rother, 2020a) follows a similar methodology to benchmark segmentation models on Cityscapes-C and PASCAL-VOC-C. (Hendrycks et al., 2019; Yun et al., 2019; Cubuk et al., 2018) improve the robustness of image recognition models to generic degradations using data augmentation. Based on the insight from (Geirhos et al., 2018) that deep neural networks trained on ImageNet seem to rely more on local texture instead of global object shape, (Kamann & Rother, 2020b) proposes a method to improve segmentation robustness to generic degradations by leveraging semantic segmentation annotations. These methods cater to generic degradations and do not focus on any specific degradation.

A similar trend is observed for depth estimation. Multiple studies (Ranftl et al., 2020; Gasperini et al., 2023; Kong et al., 2023; Sun et al., 2023) quantify and benchmark the robustness of depth estimation in the presence of degradations, especially for autonomous driving. Solutions have also been proposed to improve robustness of depth estimation (Gasperini et al., 2023; Kong et al., 2023; Sun et al., 2023) including generating synthetic data using GANs (Saunders et al., 2023) for training. Motion blur is only one of the multiple degradations they address and consequently they fail to leverage motion blur-specific insights. For ease of modeling, they synthesize blurry images by

convolving sharp images with non-linear blur kernels in $2D$ resulting in space-invariant blur. The only space variant blur they consider is 'zoom blur' (Kong et al., 2023), a radial blur that appears to radiate from (or converge toward) the center or a specific focal point of the image. This is done to simulate the blur caused when a car is moving fast along a straight road. They do not factor in depth while synthesizing such images. Moreover, since they generate $2D$ kernels to obtain space variant 'zoom blur', it becomes increasingly computationally intensive to generate kernels for each pixel of the image as the image size increases. There is no explicit modeling of localized dynamic scene blur in these works.

Motion blur is a complex degradation to generalize to due to its inherent dependency on the scene, including moving objects and occlusions, in addition to blur contributed by camera egomotion. Standard augmentation methods fail to take these intricacies into consideration. (Aakanksha & Rajagopalan, 2023) introduced class-centric space-variant motion blur in a sharp image by leveraging localization provided by segmentation annotations to model the motion blur caused by scene dynamics. To model camera egomotion, they select all the classes present in the image, which results in space-invariant camera ego-motion blur. While this simplistic modeling of camera egomotion blur is computationally advantageous, it disregards the inherent depth dependence of blur on scene geometry.

## 3 METHODOLOGY

In this work, we propose a computationally efficient augmentation strategy to introduce space-variant, depth-consistent motion blur into sharp images. An illustration of our approach is given in Fig. 1(a). We start by discussing multiple ways of generating motion-blurred images in Section 3.1. We then provide a detailed description of our Depth-consistent Motion Blur Augmentation (DMBA) strategy in Section 3.2, including the details of our $3D$ camera trajectory generation methodology.

### 3.1 REAL OR SYNTHETIC BLUR?

To the best of our knowledge, no dataset exists that provides accurate annotations for scene understanding tasks like semantic segmentation or monocular depth estimation along with blurred and sharp pairs of real images. This leaves us with two options - (i) record our own dataset or, (ii) create synthetic data.

We resort to the second option since capturing real pixel-aligned pairs of blurred and sharp images is non-trivial. Although a beam-splitter setup (Rim et al., 2020) can be used to capture pairs of sharp and blurred images of the same scene simultaneously, the precise alignment of cameras and optical elements is critical, along with careful calibration of the cameras to avoid any misalignemnts and artifacts. To be able to pair LIDAR (Geiger et al., 2013) captured sparse depth maps with the corresponding captured sharp RGB image requires accurate calibration between the sensors, which in turn necessitates carefully designed equipment and post-processing software. Importantly, already curated datasets must be leveraged for scene diversity. Hence, following existing works (Sayed &

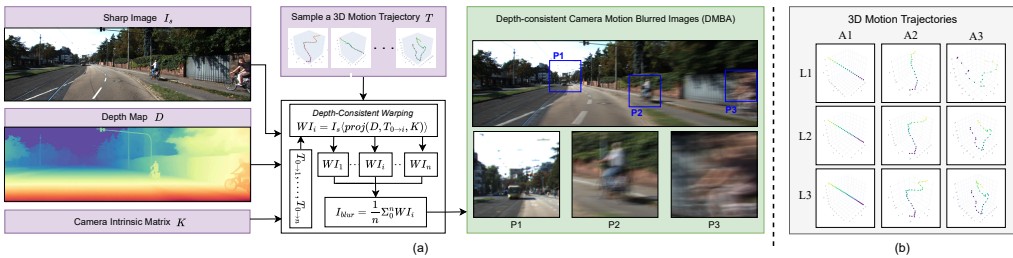

Figure 1: (a) Depth-consistent Motion Blur Augmentation (DMBA): Given a sharp image, a dense depth map, a 3D motion trajectory and camera intrinsics, we can synthesize Depth-consistent Motion Blur Augmentation (DMBA) using depth-consistent image warping. (b) Sample motion trajectories for DMBA with anxiety levels increasing from left to right (A1 - A3) for each of the three exposure levels (L1 - L3).

Brostow, 2021; Guo et al., 2019; Hendrycks & Dietterich, 2019; Kamann & Rother, 2020a; Lu et al., 2019; Zhang et al., 2020), we choose to generate synthetic data for our experiments.

Two possible approaches could be taken for synthesizing the dataset: (a) generate pseudo-ground truth supervision corresponding to real motion-blurred images by passing their sharp counterparts through a pre-trained state-of-the-art network for the required scene understanding task, or (b) synthetically blur the images for which annotations are available. For the first approach, standard deblurring datasets like GoPro (Nah et al., 2017) and REDS (Nah et al., 2019) can be used. However, these datasets were primarily designed for deblurring tasks and consequently focus on texture in the captured images and not the diversity of visible objects or depth. This makes their distributions significantly different from the distributions of standard scene understanding datasets, which renders them a poor choice to train on.

The second approach involves synthetically blurring standard datasets like Cityscapes (Cordts et al., 2016) for segmentation, and KITTI (Geiger et al., 2013) for depth estimation, in order to obtain blursharp pairs for supervision. Synthetic blur can be introduced in images in multiple ways. While Generative Adversarial Networks (GANs) have recently been used (Zhang et al., 2020; Li et al., 2020), they do not provide sufficient control and interpretability over motion and blur. Such a trained model would also fail to introduce depth-consistent blur in case the scene depth variations are significant. Other works resort to a simpler approach and blur images by convolving them with blur kernels (Sayed & Brostow, 2021). A key issue with this approach is that it generates only space-invariant blurring which corresponds to only a small subset of the entire spectrum of motion blur. It can be considered to be realistic camera-shake motion blur only if the distance between the camera and the scene is large.

## 3.2 Depth-consistent Motion Blur Augmentation (DMBA)

To introduce space-variant motion blur in a realistic manner for data augmentation, it is necessary to ensure depth-consistent blur that accurately reflects camera egomotion, particularly in scenes exhibiting substantial depth variations, which are commonly encountered in real-world settings. Since we seek to improve robustness by using large amounts of blurry images to augment training data, our approach needs to be efficient while ensuring depth-consistent blurring.

We develop our methodology grounded in $3D$ and refer to it as *Depth-consistent Motion Blur Augmentation* (DMBA). Our approach is able to generate depth-consistent motion-blurred images from sharp images in an efficient and automated manner. It hinges on three components: a) dense depth map, b) $3D$ motion trajectory, and c) camera intrinsics for the sharp dataset when available.

### 3.2.1 Generating 3D motion trajectories

Existing works on camera trajectory generation focus on generating realistic video trajectories that contain significant motion between frames. In contrast, our objective is to generate small amounts of realistic motion that may occur during the camera exposure time. We refer to our generated camera trajectories as motion trajectories and discuss their generation next.

A camera has 6 degrees of freedom (DoF) and to model camera shake, ideally, we would like to generate the motion trajectories with full DoF. Assuming the $Z$ axis aligned with the camera optical axis, motion due to camera shake is predominant only in the XY plane (Gavant et al., 2011). For small rotations, such as the motion undergone during the camera exposure time, camera rotations can be approximated as translations. Hence, we can model camera shake blur using motion trajectories having translation motion along the X and Y axes similar to (Aakanksha & Rajagopalan, 2023). However, consider the case of motion blur arising from a camera mounted on a moving vehicle. In such cases, the motion-bur can be attributed not only to camera motion caused due to hand shake but also from the forward motion of the vehicle. To include such cases, we allow translations along the Z axis in our motion trajectories.

Building on (Boracchi & Foi, 2012), we generate $3D$ motion trajectories as a straightforward extension from $2D$. To generate a $3D$ trajectory, the starting position $\overrightarrow{x_0}$ is initialized as $\overrightarrow{0}$ and we randomly sample a starting velocity from a unit sphere. Given a pose, $\overrightarrow{x_t}$, and velocity $\overrightarrow{v_t}$, the next

Figure 2: Comparison of DMBA which is space-variant, with space-invariant blur and the corresponding sharp image. In the top row, we show sharp image (left), space-invariant blurred image (middle), and image corresponding to the highest blur severity using DMBA (right). Cropped regions at different scene-depths are included in the bottom row for better visibility of depth-consistent blur. Distance from the camera decreases as we move from P1 to P3, and consequently, the blur also increases for DMBA. In contrast, space-invariant blurring fails to capture this as the blur severity remains the same across all three patches, irrespective of depth.

pose, $\overrightarrow{x_{t+1}}$, in our camera motion trajectory is given by the following equation.

$$\overrightarrow{x_{t+1}} = \overrightarrow{x_t} + \overrightarrow{v_t} + A_i \cdot (\alpha \cdot \overrightarrow{\Delta v_g} - \beta \cdot \overrightarrow{x_t}) + 2 \cdot A_i \cdot |\overrightarrow{v_t}| \cdot \overrightarrow{\Delta v_j} \tag{1}$$

where '$A_i$' denotes the anxiety level, a scalar quantity modelling jerk and velocity changes in the trajectory, $\Delta v_g$ is random acceleration drawn from $N(0, \sigma^2)$ and $\Delta v_j$ models the jerk and is randomly sampled from a sphere.

Since our work deals primarily with motion blur in the outdoors, the motion trajectories are designed accordingly. However, there is no limitation on the type of motion trajectories to be used and a full 6DoF trajectory can be used if the dataset so requires. In our experiments, we consider 3 anxiety levels for diversity and model motion blur by varying the number of camera poses being considered. We define our blur levels as $L1$, $L2$ and $L3$ corresponding to different number of camera poses considered for our experiments with $L3$ having the highest blur severity. Some examples of our generated trajectories are given in Fig. 1(b). Having generated our motion trajectories, we now discuss dense depth generation from sharp images and depth-aware warping.

### 3.2.2 DEPTH-CONSISTENT MOTION BLUR

The depth groundtruth maps provided in standard outdoor monocular depth estimation datasets are sparse and hence cannot be directly used. We use Depth Anything V2 (Yang et al., 2024), a SoTA model for dense depth estimation, to obtain the relative depth maps corresponding to each sharp image $I_s$ in the dataset. These relative depth maps are then scaled to an appropriate dataset-specific predefined range to get the metric depth $D$ corresponding to the scene.

Given a sharp image $I_s$, its corresponding sharp dense depth map $D$ and a randomly sampled motion trajectory $T$, the depth-consistent camera egomotion blur is synthesized by warping and aggregation using the following equation.

$$I_{\text{blur}} = \frac{1}{n} \sum_{i=0}^{n-1} I_s \langle \text{proj}(D, T_{0 \to i}, K) \rangle \tag{2}$$

Here, $K$ is the camera intrinsic matrix, proj(.) gives the resulting $2D$ coordinates of the projected depths of $D$ in the intermediate warped sharp image, and $\langle . \rangle$ denotes the sampling operator. Bilinear sampling is used to sample the sharp image, to get the pixel intensities in the intermediate warped images. We average '$n$' sharp warped intermediate frames to obtain the blurred image $I_{\text{blur}}$. Note that this gives us the unique ability to interpolate between frames at any desired granularity. This ensures smooth observed blur and removes the limitation arising due to camera frame capture rate for creating blurring datasets by averaging frames. An example of a blurred image using DMBA is shown in Fig.2 along with its sharp, and space-invariant blurred counterpart to highlight the depth-consistent nature of DMBA.

---

**Algorithm 1** Depth-consistent Motion Blur Augmentation (DMBA)

---

**Input:** A tuple of sharp image $I_s$, its corresponding dense depth map $D$ and camera intrinsics $K$
**Output:** $I_{\text{blur}}$
1: Sample a camera trajectory $T$ with $n$ poses
2: **for** $i = 1$ to $n$ **do**
3: $\quad$ $I_i = I_s \langle \text{proj}(D, T_{0 \to i}, K) \rangle$
4: **end for**
5: $I_{blur} = \frac{1}{n} \sum_{i=0}^{n-1} I_i$
6: **return** $I_{\text{blur}}$

---

## 4 EXPERIMENTS

We now establish through experiments the effectiveness of augmenting sharp images with our synthetically generated data in increasing robustness to motion blur. Although we show experiments only on semantic segmentation and monocular depth estimation, these are representative scene understanding tasks. Our motion blur generation strategy is generic and can be used for other scene understanding tasks as well provided dense depth maps can be obtained for them. We show that performance gains can be achieved using our augmentation scheme for both these tasks in the presence of motion blur. In Section 4.1, we discuss the implementation details followed by quantitative and qualitative results in Section 4.2. Ablations are provided in supplementary.

In this section, we show effectiveness of our approach in generating depth consistent motion blur by showing improved robustness to motion blur for segmentation and monocular depth estimation as representatives of generic scene understanding tasks and perform extensive experiments on them.

### 4.1 IMPLEMENTATION DETAILS

**Datasets:** We use publicly available outdoor autonomous driving datasets, namely, Cityscapes (Cordts et al., 2016) for our semantic segmentation experiments, and KITTI (Geiger et al., 2013) for monocular depth estimation. For evaluation, the predicted depth is restricted to the range of 0-80m, as is common practice. These datasets were specifically chosen for their considerable depth variations, which makes them a good choice for investigating the interaction between depth and blur.

**Networks:** We select DeepLabV3+ (Chen et al., 2018) and Segformer (Xie et al., 2021) as networks for our segmentation experiments owing to their competitive performance and as representatives of their specific kinds of architectures. DeepLabV3+ (Chen et al., 2018) is a standard convolutional semantic segmentation model. It includes an atrous spatial pyramid pooling module which improved segmentation performance over previous works. Segformer (Xie et al., 2021) is a recent framework which unifies vision transformer backbones with lightweight multilayer perceptron decoders to perform semantic segmentation and achieves state-of-the-art performance across datasets. For depth estimation, we use LiteMono (Zhang et al., 2023) as a representative of state-of-the-art self-supervised depth estimation network. It is a recent lightweight CNN-Transformer hybrid network making it a feasible choice for possible deployment on user-end devices.

Note that we illustrate the applicability of our approach not only to fully supervised task of semantic segmentation but also to self-supervised learning paradigm in the case of monocular depth estimation with LiteMono. During supervised training of a scene understanding task, the loss imposed with explicit supervision forces the network to learn the correct sharp output given blurry or sharp images. In contrast, for a self-supervised training paradigm, the loss driving the learning is indirect. In particular, for LiteMono, the blurry images are only shown to the depth estimation subnetwork during training and not to the pose network. The pose network receives the corresponding sharp frame and an adjacent sharp frame. Considering that blur suppresses textures and features, only sharp images are shown to the pose estimation network. During inference only the depth estimation network is needed in LiteMono. Such a training setup encourages the depth network to produce sharp, consistent depth maps for both sharp and blurry inputs.

**Comparison Baselines:** We compare our results with baselines proposed in (Aakanksha & Rajagopalan, 2023) as well as the augmentation strategy, Class-Centric Motion Blur Augmentation (CCMBA), proposed in their work. We quantify the performance gains achieved by DMBA in terms

Table 1: Comparison of mIoU values with baseline methods on the Cityscapes dataset for DeepLabV3+ model.

| Image Type | Pretrained | Finetuning | MBA | RandMBA | CCMBA | DMBA | CCMBA+DMBA |
|---|---|---|---|---|---|---|---|
| Sharp | 75.24 | 69.31 | 73.71 | 75.25 | 75.59 | **75.89** | 75.45 |
| L1 | 66.78 | 70.79 | 72.26 | 72.89 | 73.31 | **74.99** | 74.57 |
| L2 | 63.57 | 70.05 | 72.16 | 71.70 | 72.52 | 74.09 | **74.32** |
| L3 | 59.63 | 68.25 | 70.61 | 70.14 | 70.86 | **73.53** | 73.12 |

Table 2: Comparison of mIoU values with baseline methods on the Cityscapes dataset for Segformer model for depth-consistent blurring.

| Image Type | Pretrained | Finetuning | MBA | RandMBA | CCMBA | DMBA | CCMBA+DMBA |
|---|---|---|---|---|---|---|---|
| Sharp | 76.2 | 72.83 | 75.68 | 76.3 | 76.21 | 76.22 | **76.66** |
| L1 | 71.39 | 72.48 | 74.01 | 75.12 | 74.92 | 75.6 | **75.79** |
| L2 | 69.08 | 71.71 | 73.52 | 74.48 | 74.5 | 75.55 | **75.66** |
| L3 | 66.5 | 70.58 | 72.58 | 73.27 | 73.51 | 75 | **75.23** |

of robustness to depth consistent motion blur. Our first baseline is 'Pretrained' where we consider a model trained on sharp images and quantify its robustness to different levels of depth-consistent blur. The second baseline we compare with is 'Finetuning' where we make the network adapt to newer blurred images while retaining performance on previously seen sharp images. Towards this end, we first generate a set of $2D$ blur kernels following (Aakanksha & Rajagopalan, 2023). These are then used to blur each image in the training set. The same set of blurred images are used for finetuning the model pre-trained on sharp images with a reduced learning rate. Our third baseline is 'Motion Blur Augmentation' (MBA) where we seek to achieve robustness to motion blur by training (from scratch) the network with both sharp and space-invariantly blurred images. Each sample is convolved with a randomly selected blur kernel with probability $p$ during training introducing space-invariant motion blur across the entire image. This augmentation scheme does not take into account scene dynamics. We also introduce a new baseline called 'Random-Blur Augmentation' (RandMBA) where we divide each image into a '$P$' non-overlapping patches and a random subset of patches is convolved with a $2D$ blur kernel to introduce systematic space-variant blur in sharp images. In our experiments, a patch size of $W/6 \times H/3$ is considered for Cityscapes and KITTI, where $W$ and $H$ are the width and height of the image, respectively. Finally, we also compare with CCMBA which was proposed in (Aakanksha & Rajagopalan, 2023) for semantic segmentation. Since our augmentation strategy DMBA primarily models camera egomotion blur, we also employ CCMBA, to additionally introduce space-variant dynamic scene blur. The results for the combined setting are reported under 'CCMBA+DMBA' in Tables 1 and 2. For monocular depth estimation, we only compare with baselines since KITTI dataset has very limited ground truth semantic segmentation annotations which is a prerequisite for CCMBA.

We train DeepLabv3+ and Segformer networks on Cityscapes dataset, and LiteMono network on the KITTI dataset. The evaluation metric used for semantic segmentation in our experiments is standard mean-Intersection-over-Union (mIoU). For monocular depth estimation, we report standard metrics of absolute relative error (Abs. Rel.), root mean square errors (RMSE), and delta metric $\delta 1$.

In all our experiments, we use ResNet-50 as the backbone for DeepLabv3+, MiT-B0 as the backbone for Segformer, and the 'Base' configuration for LiteMono. For DeepLabv3+, a batch size of 16 with crop size of $768 \times 768$ is used for Cityscapes while for Segformer on Cityscapes, we consider a batch size of 16 with crop size of $1024 \times 1024$. For LiteMono, a batch size of 12 with image resolution of $320 \times 1024$ is used for training. In all our experiments, we use ResNet-50 as backbone for DeepLabv3+ and MiT-B0 as backbone for Segfomer unless specified otherwise. All the other hyper-parameters and training setup are the same as those in the original papers. The DeepLabV3+ and LiteMono models were trained on a single NVIDIA GeForce RTX 3090 Ti node each and Segformer models were trained using two such nodes.

## 4.2 RESULTS

To establish the increased robustness of models trained with our augmentation, we compare quantitative results with a set of baseline methods and report the results in Tables 1 and 2 for semantic

Table 3: Quantitative comparisons with baselines on LiteMono model for the KITTI dataset under different motion blur types.

| Method | Sharp | | | L1 | | | L2 | | | L3 | | |
|---|---|---|---|---|---|---|---|---|---|---|---|---|
| | Abs.Rel | RMSE | $\delta < 1.25$ | Abs.Rel | RMSE | $\delta < 1.25$ | Abs.Rel | RMSE | $\delta < 1.25$ | Abs.Rel | RMSE | $\delta < 1.25$ |
| Pretrained | **0.107** | **4.56** | 0.886 | 0.113 | 4.879 | 0.873 | 0.124 | 5.403 | 0.846 | 0.149 | 6.29 | 0.785 |
| Finetuning | 0.112 | 4.677 | 0.875 | 0.122 | 4.765 | 0.857 | 0.123 | 4.796 | 0.853 | 0.127 | **4.829** | 0.846 |
| MBA | 0.112 | 4.845 | 0.889 | 0.114 | 4.896 | 0.886 | 0.115 | 4.93 | **0.884** | 0.118 | 5.035 | **0.878** |
| RandMBA | 0.109 | 4.782 | 0.891 | 0.112 | 4.933 | 0.885 | 0.114 | 4.962 | 0.882 | 0.118 | 5.065 | 0.875 |
| DMBA | 0.108 | 4.692 | **0.893** | **0.11** | **4.727** | **0.887** | **0.112** | **4.794** | 0.882 | 0.117 | 5.025 | 0.871 |

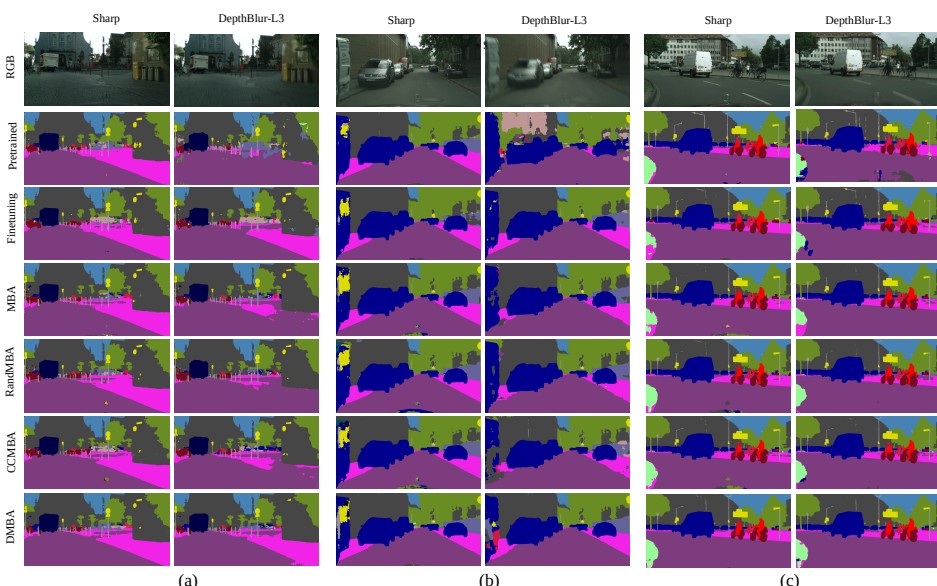

(a) (b) (c)

Figure 3: Qualitative comparisons with baselines and CCMBA for sharp and motion-blurred images using DeepLabV3+ on Cityscapes dataset is shown in (a), (b) and (c).

segmentation, and Table 3 for depth estimation. We show qualitative comparisons for semantic segmentation in Fig. 3 and depth estimation in Fig. 4. Since no works exist that investigate the robustness to real dynamic scene motion blur, we compare our qualitative results obtained for blurred and sharp images from GoPro and REDS datasets with the baseline methods for both the tasks.

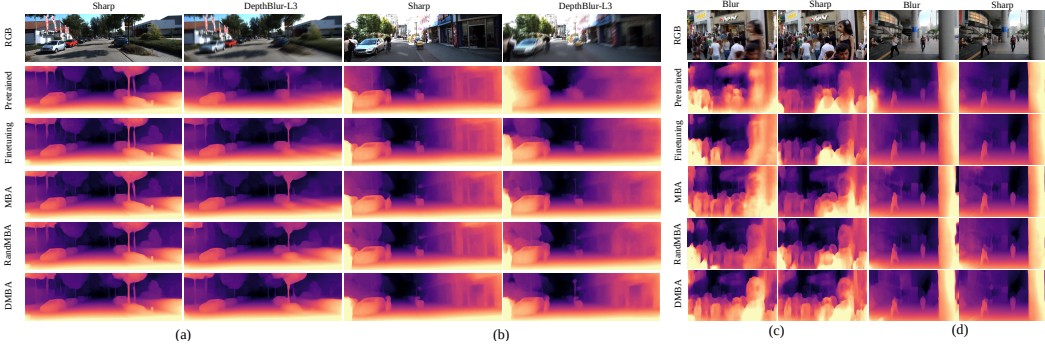

(a) (b) (c) (d)

Figure 4: Qualitative comparisons with baselines for sharp and motion-blurred images using Lite-Mono on KITTI dataset is shown in (a) and (b) and comparisons with baselines for space-varying real blur on GoPro and REDS dataset is shown in (c) and (d). Note that our method DMBA captures finer details better than all baselines, especially at close distances.

### 4.2.1 Quantitative Results

To ensure consistency across experiments and to facilitate fair assessment of robustness, we construct fixed test sets for the depth-consistent camera motion blur which is synthesized using DMBA strategy. All these images are generated and saved in advance for reproducibility.

We provide comprehensive comparisons with the baseline methods in Table 1 and 2 for DeepLabV3+ and Segformer, respectively, on the Cityscapes dataset. A general trend in performance is observed across all experiments. The 'Pretrained' baseline performance drops severely as we move from blur levels L1 to L3. The 'Finetuning' baseline improves the performance across blur levels L1, L2 and L3 but at the cost of performance drop for sharp images, which is not desirable. The 'MBA' and 'RandMBA' baselines show improved performance across all levels of blur while retaining performance for clean sharp images.

For Cityscapes dataset and using DeepLabV3+, our approach DMBA achieves 2.29%, 2.48% and 3.77% performance gains for L1, L2 and L3 levels of depth-consistent blurring over the best performing baselines while retaining performance on sharp images. For Segformer, also, we see a similar trend. DMBA performs better than the next best performing baseline by 0.89%, 1.56% and 2.34%, respectively, for L1, L2 and L3 blur levels. Note that for both the architechtures, the best performance is achieved by 'DMBA' or 'CCMBA+DMBA' which highlights the efficacy of our approach.

For self-supervised monocular depth estimation, quantitative results are provided in Table 3. DMBA performs most consistently across all the three metrics reported across all blur levels as well as sharp images. This establishes that performance gains are achieved across all the blur types consistently and across different tasks, when using our DMBA strategy.

### 4.2.2 Qualitative Results

Qualitative comparisons with baseline methods on Cityscapes and KITTI datasets are given in Fig. 3 and Fig. 4. In Fig. 3 DMBA is able to segment out the footpath (in pink) and the foliage (in green) in foreground more consistently than other baselines. In particular, for Fig.3(c), the foreground region is segmented better by DMBA compared to baselines. Additionally, on zooming in, our results suffer from lesser artefacts across both images. Our results are clearly more consistent across both sharp and blur images. For depth estimation, our approach is able to pick up details nearer to the camera, like the small shrub on the footpath in Fig. 4(a) which is visible on zooming in, even in the presence of highest level of blur. Similarly, in Fig. 4(b), the silhouette of the person in the foreground is captured more consistently across both sharp and blurred images. For space-varying real blur, a sample image from GoPro and REDS dataset each is shown in Fig.4(c) and Fig. 4(d). Our results are clearly more consistent across blurry and sharp images, particularly for objects nearer to the camera.

**Supplementary** In our supplementary material, we include an ablation study on the effect of sampling blurred image with varying probabilities, along with a discussion on the applicability of our approach to datasets where camera intrinsics are unavailable or depth variations are not significant within the scene. We also include higher-resolution images for better visualization.

## 5 Conclusion

In this work, we improve the robustness of scene understanding tasks such as semantic segmentation and depth estimation in the presence of realistic camera egomotion blur. We develop a comprehensive depth-consistent motion blur augmentation strategy to model space-varying blur in an efficient and realistic manner. We propose an approach for depth-consistent motion blurring leveraging depth foundation models to warp and aggregate images for modeling camera egomotion blur. Our proposed method is task agnostic and can be applied to any scene understanding task. Experiments across two different representative scene understanding tasks and training paradigms demonstrate the effectiveness of our approach and its applicability to improving the robustness to realistic motion blur without increasing model parameters or inference time.

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
