# Depth-consistent Motion Blur Augmentation
# SUPPLEMENTARY MATERIAL

In this supplementary material, we include the following.

1. Ablation studies on the Cityscapes dataset for the segmentation task investigating the effect of using different probabilities for sampling the blurred data.

2. Discussion on the choice of camera intrinsics and results on PASCAL VOC dataset.

3. Additional qualitative results for segmentation and depth estimation.

All the tables, figures and sections in this supplementary are numbered starting with an 'S' to make clear distinctions between the contents in the main paper and supplementary.

## S1. Ablation Studies

In this section, we document the results for ablations studies on our DMBA strategy. The experiments are conducted on the Cityscapes dataset for DeepLabV3+ for semantic segmentation, without loss of generalizability. To study the impact of sampling probability of blurry images on training we train all the models were trained for 60000 iterations with a batch size of 12. As can be seen from Table S1, increasing the probability of an image being blurred makes the model learn slightly faster for blurry images while keeping a lower probability ensures faster training for sharp images.

Table S1: Ablations on the effect of probability of blurring on performance-mobilenet

| Probability of Blurring | mIoU | | | |
|:---:|:---:|:---:|:---:|:---:|
| | Clean | L1 | L2 | L3 |
| p=0.3 | 73.11 | 71.54 | 70.65 | 69.42 |
| p=0.5 | 72.53 | 71.07 | 70.04 | 68.85 |
| p=0.7 | 72.55 | 71.41 | 70.49 | 69.68 |

## S2. Discussion on Camera Intrinsics and PASCAL VOC

Depth-consistent camera motion blur is likely to have a more significant impact on the Cityscapes dataset compared to the Pascal VOC dataset. Cityscapes consists of street-level driving scenes captured with onboard cameras, where there are considerable depth variations between foreground objects (such as cars, pedestrians, and traffic signs) and background elements (such as buildings, roads, and the sky). In these environments, camera motion creates spatially varying blur due to parallax effects, making depth-consistent modeling essential for realistic data augmentation and for preserving scene geometry in tasks like semantic segmentation and depth estimation.

In contrast, the Pascal VOC dataset mainly includes internet photographs that exhibit limited and often shallow depth variations. As a result, blur artifacts in this dataset tend to be more uniform and less influenced by the scene's geometry. Therefore, while uniform motion blur may replicate some distortions in the Pascal VOC dataset, depth-consistent motion blur more accurately reflects the challenges found in real-world scenarios similar to those in the Cityscapes dataset.

We used the provided camera intrinsics matrix, $K$, for the Cityscapes and the KITTI dataset. For the PASCAL VOC dataset, we assume the camera intrinsic matrix to be $K = \begin{bmatrix} f \times w & 0 & w/2 \\ 0 & f \times w & h/2 \\ 0 & 0 & 1 \end{bmatrix}$, where $f = 1$ is the focal scale, $w$ is image width and $h$ is image height. A camera intrinsic matrix, $K$, defined this way is commonly used as a standard initialization in COLMAP and related structure-from-motion pipelines.

Table S2: Comparison of mIoU values with baseline methods on the PASCAL VOC dataset for DeepLabV3+ model. Since PASCAL VOC contains very limited depth-variations within an image, the performance gains achieved by DMBA are not as significant as for datasets like Cityscapes.

| Image Type | Pretrained | Finetuning | MBA | RandMBA | CCMBA | DMBA |
|---|---|---|---|---|---|---|
| Clean | 74.35 | 58.52 | 75.26 | 75.01 | 75.25 | 74.00 |
| L1 | 70.95 | 67.08 | 71.19 | 73.70 | 74.92 | 73.22 |
| L2 | 62.11 | 66.01 | 69.04 | 70.67 | 73.40 | 72.10 |
| L3 | 43.98 | 60.73 | 62.03 | 63.87 | 67.65 | 68.28 |

## S3. Additional Qualitative Results

We provide additional results as well as the results shown in the main paper at higher resolution here for better visualization. In particular, in Fig.S1-S4 we include the results already shared in the main paper at a higher resolutions with Fig.S1 corresponding to semantic segmentation on Cityscapes with DeeplabV3+ and the remaining pertaining to LiteMono. To further emphasise the superior generalizability of our approach, we include additional results on the REDS and GOPRO dataset for self-supervised depth estimation in Fig.S5. We highlight the consistency between our results for blur and sharp images and also include zoomed and cropped regions adjacent to the images for better visualization.

Figure S1: Qualitative comparisons with baselines for semantic segmentation on Cityscapes dataset.

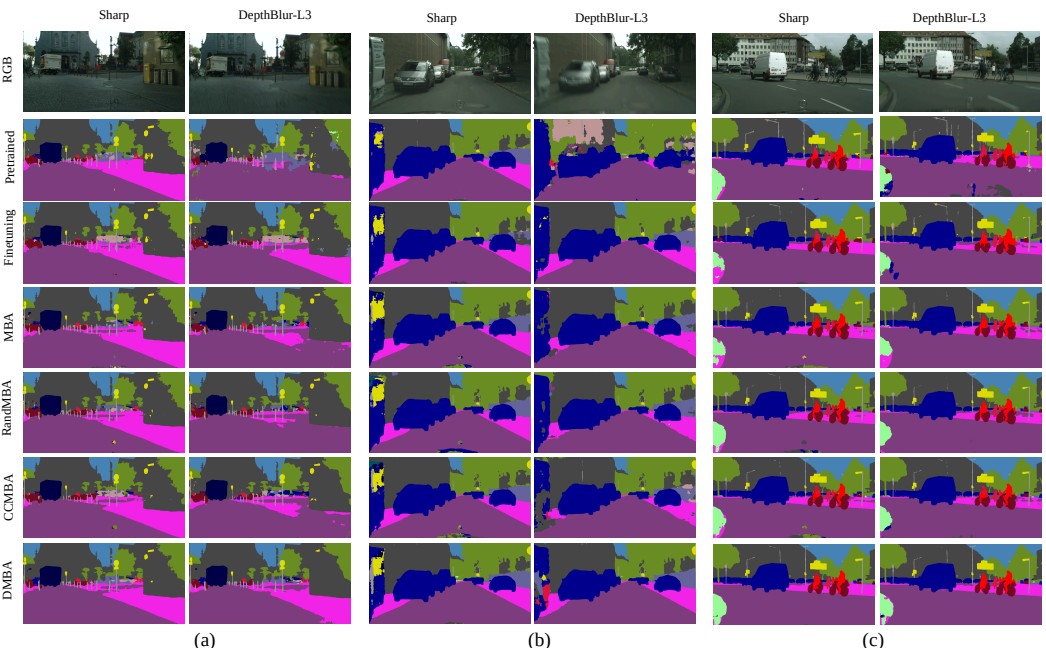

Figure S2: Qualitative comparisons with baselines for depth estimation on sample image 1 from KITTI.

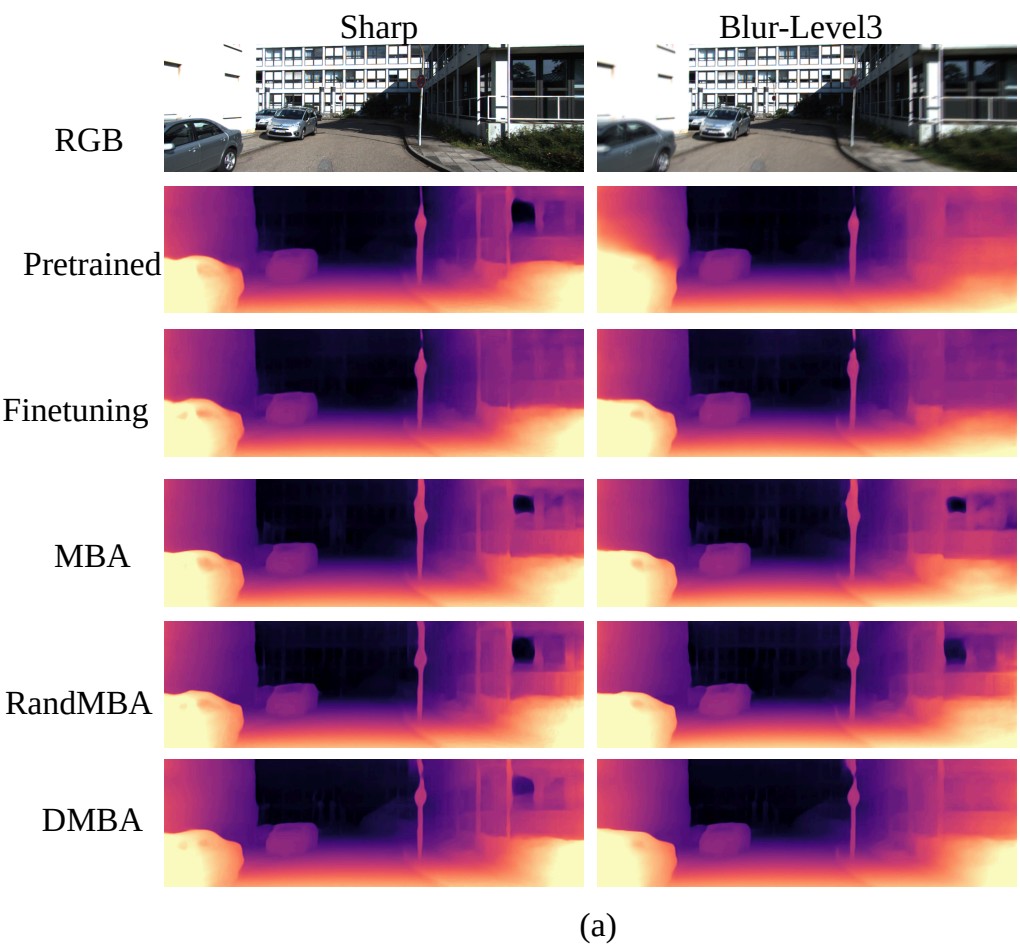

(a)

Figure S3: Qualitative comparisons with baselines for depth estimation on sample image 2 from KITTI.

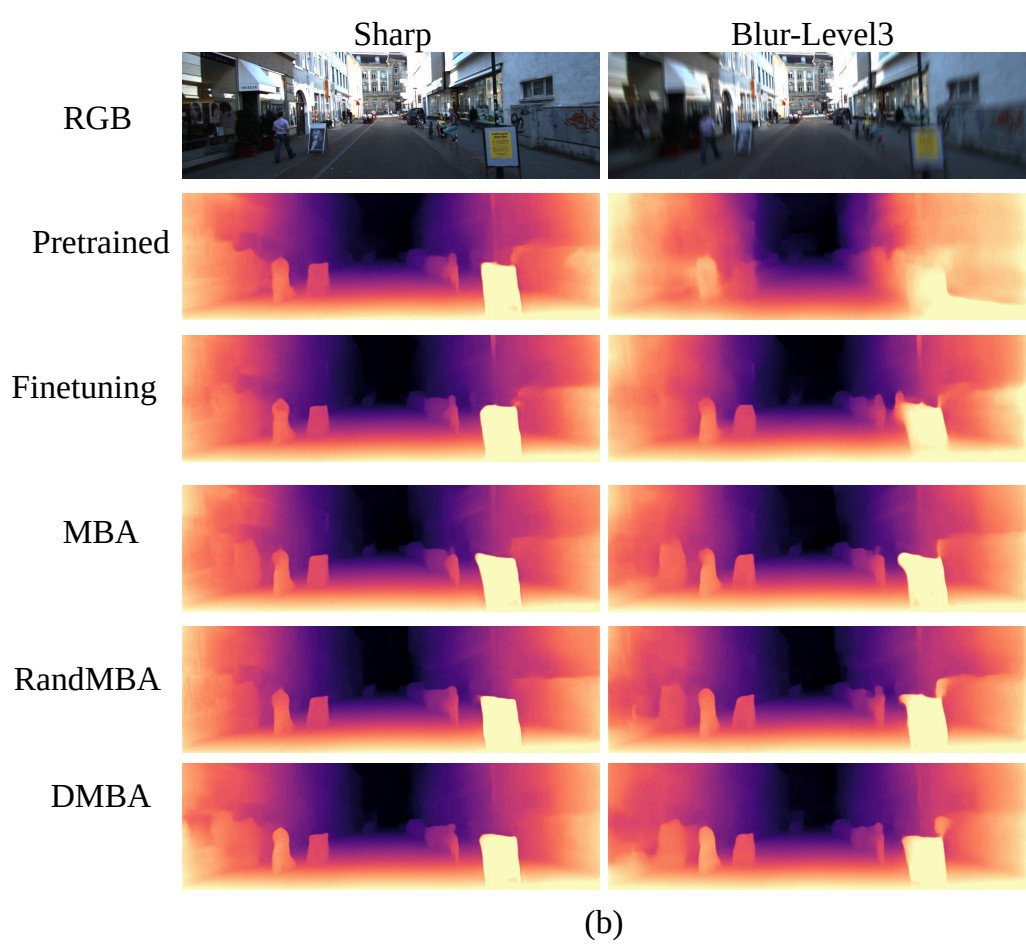

(b)

Figure S4: Qualitative comparisons with baselines for depth estimation on sample image 3 from KITTI.


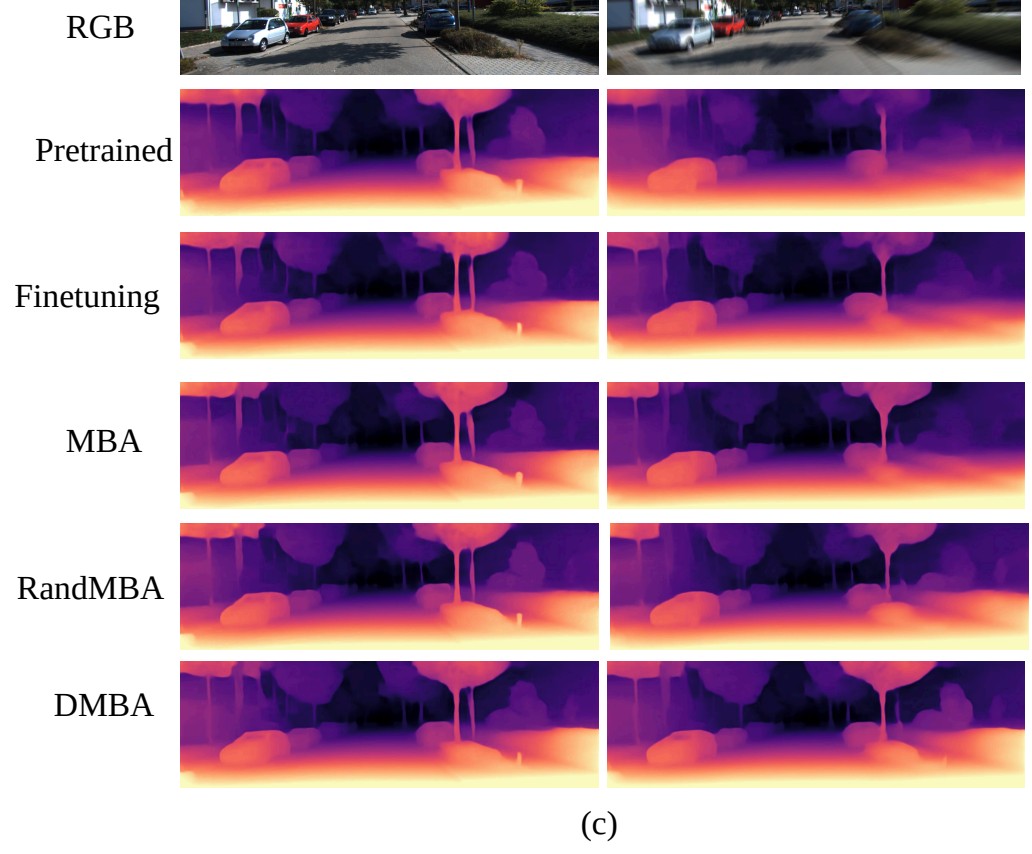

(c)

Figure S5: Qualitative comparisons with baselines for depth estimation on the REDS and GOPRO datasets.


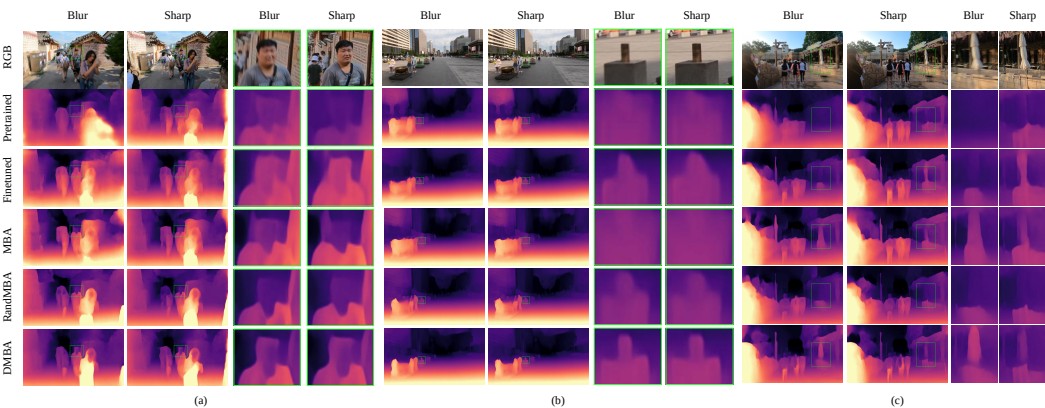