# OpenReview forum: "Depth-consistent Motion Blur Augmentation"
_ICLR.cc/2026/Conference — ICLR 2026 Conference Withdrawn Submission_

### Official Review · Reviewer_YMgh · 2025-10-23

**Soundness:** 2
**Presentation:** 2
**Contribution:** 2
**Rating:** 2
**Confidence:** 5

**Summary:**

This paper proposes the Depth-consistent Motion Blur Augmentation (DMBA) to model the motion blur. It consists of two parts: the 3D motion trajectory generation and the depth-consistent motion blur. Experiments on two tasks demonstrate the effectiveness of the proposed method.

**Strengths:**

This paper proposes a 3D trajectory generation pipeline guided by depth to model the motion blur, which involves 6 DoF to simulate camera motion. The experiments show the superiority of the proposed method.

**Weaknesses:**

1. The key ideas of the proposed method lie in the 3D motion trajectory and the depth estimated by the Depth Anything model. However, the 3D motion is not new and is widely discussed in traditional computer graphics. Besides, the depth is predicted by the existing pretrained model. Thus, the paper seems to lack novelty. A clarification of novelty is encouraged.
2. The introduced "depth-consistent" is confusing. What is the role of depth in the method? And, in Eq. 2, the "proj(.)" involves depth as an input. Both the detail and the motivation are missing.
3. Sec. 3.2.2 notes that the existing depth groundtruth maps are sparse and the Depth Anything model provides better results. An ablation study is supposed to validate this point, yet it is not included in the current paper.

**Questions:**

1. The method seems to overlook the local motion of objects. How can it handle such scenes with local motions?

---

### Official Review · Reviewer_9jTg · 2025-10-28

**Soundness:** 3
**Presentation:** 3
**Contribution:** 2
**Rating:** 0
**Confidence:** 5

**Summary:**

This paper proposes to generate realistic motion blurred images by considering the different depth of different pixels, thus the generated images are with realistic camera motion and depth consistency.

**Strengths:**

The reported experiment results are marginally improved.

**Weaknesses:**

1. The contribution is very limited. The conventional methods generate the motion blurred images by averaging a sequence of images in a video, which is efficient, motion realistic, and depth consistent. The proposed method actually perform the same thing.

2. The explanation of Eq.(1) is missed and confusing. What is the physical meaning for multiplying jerk and velocity? How about the subtraction between the acceleration and pose?

3. There is no technique contribution from this work.

**Questions:**

Please refer to the Weaknesses.

---

### Official Review · Reviewer_eTca · 2025-10-30

**Soundness:** 3
**Presentation:** 2
**Contribution:** 3
**Rating:** 4
**Confidence:** 2

**Summary:**

This paper introduces Depth-consistent Motion Blur Augmentation (DMBA), that leverages depth foundation models to generate realistic, space-variant, depth-aware motion blur for modeling camera egomotion. Unlike prior space-invariant approaches, DMBA produces motion blur that aligns with scene geometry, thereby enhancing robustness in semantic segmentation and monocular depth estimation tasks.

**Strengths:**

1. Performance: The proposed DMBA is not only effective on its own but also demonstrates strong complementarity with existing augmentation methods such as CCMBA, leading to further performance improvements when the two are combined. The approach is conceptually clear, computationally efficient, and broadly applicable across vision tasks requiring robustness to motion blur.

**Weaknesses:**

1. Deformation Blur: The method primarily handles motion blur induced by camera or rigid-object motion, without accounting for non-rigid deformations (e.g., folding fingers, facial movements) where object shape changes over time. As a result, DMBA may fail to model blur from deformable or articulated motion.

2. Reference: Some related works are missing from the discussion. Since the main contribution involves using 3D trajectories informed by depth, references such as * should be cited and compared.

*Controllable Blur Data Augmentation Using 3D-Aware Motion Estimation (ICLR 2025)
GS-Blur: A 3D Scene-Based Dataset for Realistic Image Deblurring (NeurIPS 2025)

**Questions:**

1. The number of sharp frames (n) used to synthesize motion blur may play a crucial role in realism and robustness. Could the authors include an ablation study exploring how varying n affects performance and the fidelity of generated blur?

---

### Official Review · Reviewer_pw8S · 2025-11-01

**Soundness:** 2
**Presentation:** 3
**Contribution:** 2
**Rating:** 4
**Confidence:** 4

**Summary:**

The paper proposes a method to augment standard vision datasets (e.g. for segmentation or monocular depth estimation) to include synthetic camera blur, The standard training datasets are augmented using the proposed strategy, and the results show that the achieved scores outperform standard settings, Improved performance is also shown on real-world datasets, such as GoPro and REDS.

**Strengths:**

* The paper is well-written and the ideas are sound.
* The proposed strategy indeed shows slightly improved scores on several benchmark datasets.

**Weaknesses:**

* Motion blur is considered only for camera motion, but not for object motion or for changing intrinsics blur.
* It's not specified how the depth maps are scaled to metric depth.
* Some works that are mentioned as recent are from 2021, which is in terms of deep learning work not so recent.

**Questions:**

Would it be possible to also add other types of motion blur?

---

### Note · Authors · 2025-11-13

I have read and agree with the venue's withdrawal policy on behalf of myself and my co-authors.